# The Impact of High-Speed Rail on Economic Development: A County-Level Analysis

Fangting Chi [1] and Haoying Han [1,2,*]

1    College of Civil Engineering and Architecture, Zhejiang University, Hangzhou 310058, China;
     22012149@zju.edu.cn
2    Center for Balance Architecture, Zhejiang University, Hangzhou 310058, China
*    Correspondence: hanhaoying@zju.edu.cn; Tel.: +86-136-0651-1156

**Abstract:** High-speed rail has an important impact on the location choices of enterprises and the labor force, which is reflected in a complex space–time process. Previous studies have been unable to show the change characteristics between enterprises and the labor force at the county level. Therefore, based on the new economic geography theory, we first constructed a theoretical analysis framework to explore high-speed railway's impact on county economy development and then obtained the two economic subdivision factors' impacts: industrial enterprises and secondary labor force. Then, based on the panel data of 1791 county units in China from 2003 to 2019, the study constructed a multi-period PSM-DID model to empirically explore high-speed rail's impact on the county's agglomeration of industrial enterprises and secondary labor force. The results show that high-speed rail has a long-term negative effect on the county area's agglomeration of industrial enterprises. From the perspective of the labor force, high-speed rail has a long-term and continuous positive effect on the agglomeration of the secondary labor force in county units.

**Keywords:** high-speed rail; county territory; PSM-DID; industry; secondary labor force

## 1. Introduction

Transportation system infrastructure has historically been seen as a crucial component of regional economic development and as having a significant positive externality [1,2]. High-speed rail has grown in importance because of transportation infrastructure expansion [3]. By reducing the physical and temporal distance between cities, increasing the population, capital, information, and technology mobility, and improving the relative accessibility between cities, high-speed rail boosts the location benefits of stations and cities along the line [4,5]. According to studies, high-speed rail significantly impacts the local economy growth in terms of both industry and population [6–9]. Therefore, for regional economic integration, sustainable development, and reducing global poverty, qualitative and quantitative evaluation of the corresponding relationship between high-speed rail networks and population and industry is of great practical significance. The research on China's high-speed rail network's impact is important for planning and building the world's rail networks because China is a leading country in this field [10].

Scholars have domestically and internationally started to investigate high-speed rail networks' effects on regional economic development considering multiple factors such as land [11], immigration [12], employment [13], highly skilled labor [14], manufacturing enterprises[7], and service industry [15]. Research on high-speed train networks' regional effects in foreign literature dates back to 1967 [16]. Sands studied the regions in Japan where Shinkansen was introduced, and he concluded that high-speed rail encouraged population growth and flow [17]. Okamoto and Sato examined the Kyushu Shinkansen and concluded that opening high-speed rail lines increased land prices in metropolises to the detriment of smaller cities. According to Blum et al., regional corridor development, location restructuring of businesses and families, and economic functional zone specialization in Western

industrialized countries were all facilitated by the high-speed rail network [18]. Chen and Hall investigated how the British InterCity 125/225 affected British economic geography, and one-, two-, and above-two-hour-away metropolitan regions affected by high-speed rail were considered the three key regional layers [19]. Heuermann and Schmieder investigated how the expansion of Germany's high-speed rail network affected workers' commute choices and concluded that it led people who lived in large cities to move to smaller ones [8]. In 2011, Willigers and Van Wee et al. conducted a representative study on the location choices of corporate offices in the Netherlands and proposed that high-speed rail would affect corporate office location choice and that high-speed rail accessibility would have a significant influence on the location choice of businesses, particularly knowledge-intensive businesses [20]. According to Diao M.'s analysis of China's "four vertical and four horizontal" high-speed rail networks, businesses can relocate from megacities to second-tier cities near high-speed rail corridors thanks to intercity trade, labor mobility, and knowledge spillover [21]. The relationship between a high-speed rail network and local economic growth has also been examined by both domestic and international experts at many scales, including the European [3], national [12], urban agglomeration [6], provincial, and city levels. In conclusion, studies on high-speed rail networks' economic impacts typically focus on examining a single element, and the research scale neglects providing small- and medium-sized towns and counties the attention they require.

High-speed rail's "siphon effect" and "trickle effect" counterbalance the economic growth of urban units. The "siphon effect" describes how, following the opening of high-speed rail, the great allure of big cities is more likely to draw resources, labor force, and businesses from smaller cities along the route, thus harming the growth of small- and medium-sized cities. The term "trickle effect" refers to how larger surrounding cities aid smaller ones through consumption, employment, industry, and other factors to close the regional imbalance. Due to its impact on the supply and turnover of the professional labor market, high-speed rail has grown to be the most complicated issue in firms' location choices [9]. Most studies on how rail affects labor have concentrated on rural-to-urban migration, neglecting the "trickle effect" of high-speed rail's exodus of skilled people to neighboring counties. The high-speed rail region will increase workforce mobility and draw more businesses that require large and specialized labor forces. Obviously, the "siphon effect" has significantly and positively impacted the growth of key cities' economies. However, we cannot determine the exact effects of the "siphon effect" and "trickle effect" on economic growth and population mobility for other urban units (county-level cities and counties) that have high-speed rail based on impressions.

Here, we carry out an empirical study on 1791 county units in China and provide a theoretical framework based on new economic geography theory. China is unquestionably an innovator in high-speed trains. China's high-speed rail system, with a total length of 19,000 km, surpassed the rest of the world's network in length by 2015. China's high-speed train network continues to reach thousands of counties regardless of the country's vastness. The study on the effects of high-speed rail is typical in that it covers a large sample and a long period of time and has reference value for the transportation and economic development of small- and medium-sized cities both domestically and overseas, as well as for new urbanization and suburbanization.

This study's aims were to: (1) Investigate the interactive mechanism of the county's population and business being impacted by the high-speed rail network. (2) Analyze high-speed rail's influence on the county's secondary industries and the labor force agglomeration of industrial enterprises. (3) Answer the question, what impact does the introduction of high-speed rail have on the county's economic growth mode and development transformation?

## 2. Analysis Framework

The inverted "U"-shaped curve in the core–edge model proposed by Krugman proves that, under the interaction of increasing return to scale, population flow, and transporta-

tion cost, the forward and backward industry correlation effect is the strongest when the transportation cost is at the intermediate level [22]. The path dependence and iceberg transport cost proposed by the new economic geography theory demonstrate transportation's importance in the industrial agglomeration process. The high-speed rail network's economic impact encourages the movement and concentration of companies and staff. The high-speed rail network's impact on the county economies' evolution has received a lot of attention from the academic community because of the extensive high-speed rail network building in China's counties. The relationship between "high-speed rail and population", "high-speed rail and industry", "high-speed rail and relative accessibility", and other discrete elements or with urban spatial structure has been examined and analyzed in previous research. However, a particular regional economy's growth is the consequence of the coordinated actions of numerous complex elements, and a high-speed rail network's impact on a county economy's growth cannot be presented scientifically and precisely through a cursory investigation of a single aspect. Here, we look at the correlation between the high-speed rail network and county economic growth. Starting with the two economic components of businesses and the labor force, we investigate the mechanism of high-speed rail networks on county economic development based on new economic geography theory.

## 2.1. The Relationship between High-Speed Rail Network and County Economic Growth

County units are all part of a complex urban network system from the perspective of the overall regional spatial structure. The connections with other urban units and the comparative advantages with units of the same rank are important decisive factors for economic development. The local advantages, industrial characteristics, and economic foundation of county units will affect their subsequent economic development's quality and speed. Introducing high-speed rail widens the pathway for factor circulation, and the interaction demand for factor flow among urban units encourages the creation and development of high-speed rail which, in turn, supports and directs urban unit development and impacts the distribution trend of enterprises and labor factors. We developed a flow model of four types of urban units (high-speed urban area, non-high-speed urban area, high-speed county, and non-high-speed county) and two types of economic elements (enterprises and labor force) under the influence of high-speed rail based on this and in combination with pertinent examples in the literature (Figure 1). The focus is on investigating high-speed rail's effects on the flow of economic components of county units and on a more intuitive study of the interplay between the two elements and urban unit objects.

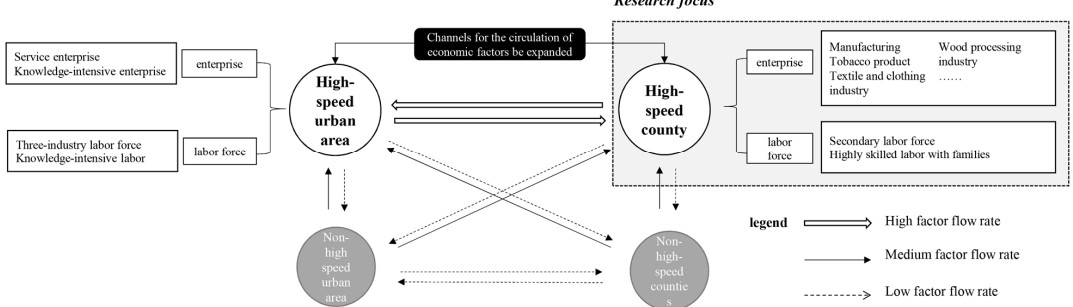

**Figure 1.** Schematic diagram of flow model of enterprise and labor force factors under high-speed rail's influence.

The relevant statistics and literature reviews demonstrate that high-speed rail broadens the production factor's circulation channel and brings it spatially and temporally closer to the center metropolis. According to the analysis of the factor flow model in Figure 1, two aspects are mostly responsible for luring businesses to the high-speed railway county. On the one hand, enterprises are forced to relocate to counties from the urban areas that high-speed rail has opened up due to the knowledge spillover effect of central cities [14] and congestion costs [7,21], particularly low-end manufacturing, textile, and other indus-

trial enterprises. Because central cities are increasingly clustered in terms of population, information and capital do not directly participate in production and modern service and knowledge-intensive sectors tend to be concentrated there [15,23]. On the other hand, the impetus comes from urban units without high-speed rail service, as the expansion of the consumer market [13], the specialized labor market [24,25], and the product input channels [26] encourage businesses to group together in high-speed rail counties. Similarly, the driving force to attract labor force in a high-speed rail county mainly comes from two aspects: on the one hand, the secondary labor force seeks employment possibilities, lowers living expenses, and considers family emigration [8,12]. Meanwhile, high-speed trains also increase the externality of human capital, and major cities are where talents, innovations, and ideas prefer to congregate [27]. On the other hand, the second and third industrial workforces from non-high-speed rail towns relocate there for various reasons, including job searching and lifestyle improvement [28].

Here, we look at the agglomeration tendency that high-speed rail has had on the two economic components of businesses and labor. Additionally, the two economic forces that are most likely to congregate in counties served by high-speed rail can be separated into industrial firms and the accompanying secondary labor force, according to the flow model study. The following research's key emphasis is the relationship between the two economic forces and high-speed rail opening.

## 2.2. Effect Mechanism of High-Speed Rail Network on County Economic Development

Exploring the mechanism between the high-speed rail network and corresponding economic growth is of long-term significance to county economies' development and transformation. As a key conduit for economic factor movement, such as inter-regional capital, information, and human flow, the high-speed rail network reconstructs the market share of factor flow. We examine the dynamic mechanism underlying the county economy's development and transformation path under the influence of high-speed railway, focusing on the two main economic segmentation factors of industrial enterprises and secondary labor force based on the findings in the previous section (Figure 2).

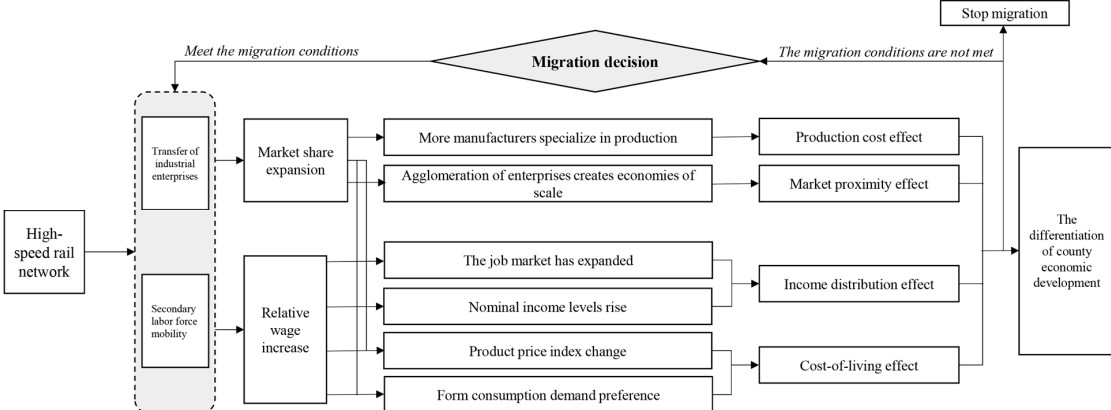

**Figure 2.** Action mechanism diagram of high-speed rail network on county economic development and transformation path.

For industrial enterprises, counties with high-speed rail may draw industrial business clusters. The market share increase enables more manufacturers to carry out more specialized production, and the grouping of businesses creates economies of scale, thus reducing production costs [29] and improving market access [30], both of which encourage industrial industry growth in counties. Regarding the secondary labor force, the high-speed rail network speeds up labor factor movement, the decline in living expenses such as housing costs, rent, consumption, and education increases relative wages, and the concentration of industrial enterprises also widens the job market and impacts income distribution [14]. The

cost of living is lowered and the cost of living effect is produced by changes in the product price index and customer demand preferences [31].

In conclusion, industrial businesses and secondary labor force distribution restructuring and agglomeration will collaborate to foster regional economy development independent of geographic and spatial considerations. Nevertheless, because of the unique characteristics of high-speed rail counties in the urban system, the "siphon effect" of central cities produced by high-speed rail may prevent industrial enterprises and the secondary labor force from congregating in high-speed rail counties, and it may even cause enterprises and populations that were originally located in high-speed rail counties to reverse to central cities. Thus, the economic variables most likely to congregate in counties opened by high-speed rail are industrial companies and the secondary labor force. The precise agglomeration trend, however, cannot be determined with sufficient accuracy based on theoretical study because of central cities' influences. Thus, high-speed rail counties must necessarily undergo reasonable economic development.

### 2.3. Exploration of Different Evolutionary Paths of High-Speed Railway Counties

Introducing high-speed rail considerably boosted the high-speed rail counties' accessibility to high-speed rail urban areas when compared to non-high-speed urban areas and counties. Regarding industrial enterprises, the cost effect of lower factor flow costs and the market proximity effect of a wider potential market scope encourage the establishment of new businesses. Consequently, the labor force demand first exceeds supply, and the county is transformed into a satellite city formed by industrial agglomeration, into an "auxiliary city". Regarding the labor force, the county will transform into a satellite city formed by population agglomeration, into a "sleeping city", because the secondary labor force supply leads the settlement of industrial enterprises due to the living cost effect caused by the increase in new service supply and the income distribution effect caused by the expansion of potential employment opportunities. The mechanism and differentiation of the impact of high-speed rail opening on high-speed rail counties' transformation and development are shown below (Figure 3).

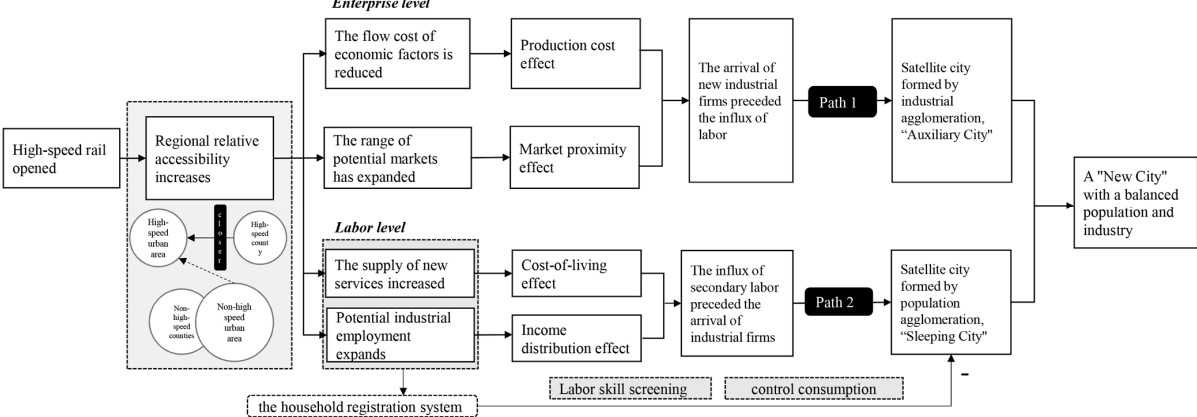

**Figure 3.** Mechanism and differentiation of high-speed railway opening affecting transformation and development of high-speed railway counties.

Several academics have also proposed that counties with high-speed rail systems attract new businesses and citizens. The core of industrial transfer is the process of business relocation and location adjustment. New businesses locate themselves in high-speed railway counties for various reasons. The primary factor is that secondary industries such as manufacturing and industry in developing nations typically originate in the major towns [32]. To save money on land and labor, businesses are relocating to nearby towns [33]. Additionally, knowledge-intensive businesses have high location and transportation needs [34] and a tendency to congregate in high-speed rail counties due to their

high technology and demand for information exchange. The primary driver for people to relocate to high-speed rail counties is the significant decrease in immigration costs, which encourages the movement of labor from relatively underdeveloped areas to these counties and the significant reduction in commute times, which increases employment opportunities and decreases living expenses [35].

*2.4. Summary of the Chapter*

To further address the research questions posed in the introduction, the chapter explores the mechanism by which the high-speed rail network affects county economic development. It also examines the agglomeration and exodus of businesses and workers in four different types of urban units (high-speed urban area, non-high-speed urban area, high-speed county, and non-high-speed county). High-speed rail opening in a county widens the channels of factor circulation, thus bringing high-speed rail counties closer to the center urban region in terms of both time and geography, according to the analysis framework. The high-speed railway significantly impacts industrial business movement and concentration as well as the secondary labor force in the county area. The original equilibrium state is disrupted by the high-speed railway's opening, and businesses and the labor force decide to migrate accordingly. It creates a fresh chance for the high-speed railway county to change and develop. We conjecture there are two primary paths for high-speed railway county transformation and growth based on analyzing enterprise and labor force levels: the first route is the formation of a satellite city formed by industrial agglomeration, the "auxiliary city"; another path is formation of a satellite city of population agglomeration and development by the influx of a large number of non-agricultural laborers, the "sleeping city".

## 3. Research Design

*3.1. Research Problem*

According to the analysis framework, from the standpoint of industrial enterprises, a high-speed rail opening greatly lowers industrial companies' transportation costs in county regions, thus affecting market access and influencing production costs. However, it also heightens the center urban area's siphon effect, which increases industrial businesses' propensities to relocate to the central city with a higher total external income. Here, we propose hypothesis 1: under the current national circumstances, a high-speed rail opening will have long-term adverse effects on industrial companies at the county level and weaken industrial enterprises' agglomeration level at the county level. Regarding the secondary labor force, a high-speed rail opening lowers living expenses and transportation costs and creates a cost-of-living effect to draw laborers into the central metropolitan region. On the other hand, the availability of new infrastructure services and the expansion of employment prospects affect income distribution and support the transformation of the agricultural population into a non-agricultural one. Here, we propose hypothesis 2: introducing high-speed rail will have a long-lasting and positive effect on the county's secondary labor force concentration. Additionally, hypothesis 3 is proposed considering the conclusion of the counties' development and transformation from the perspective of high-speed rail opening in the analysis framework 2.3: a satellite city formed by population agglomeration, "sleeping city" is formed because of the high-speed rail opening, which alters counties' economic growth styles. Industrial enterprise settlement precedes the influx of the secondary labor force.

Based on the panel data of 1791 county-level urban units (county-level cities and counties) across the nation of China from 2003 to 2019, this chapter employs the multi-period PSM-DID model to conduct the appropriate research design to test the three assumptions.

*3.2. Method Selection*

The selection bias issue with county unit samples was successfully resolved using the propensity score matching technique (PSM). Using the differential difference, the "difference-in-differences" analysis (DID) effectively examined the policy impact of high-

speed rail on the county unit while resolving the endogeneity issue engendered by the incomplete dependent variables of the county unit itself during the research process. The individual differences between the various locations prior to the high-speed rail inauguration will be disregarded by using the cross-section model and the time effect will be disregarded by using the time series model to avoid the constraints of the single difference technique. To evaluate high-speed rail's effect on county economic development, we primarily draw on the methodological work of Goodman-Bacon (2021) [36] and Callaway and Sant 'Anna (2022) [37] and uses of the multi-time point differential approach (DID, differences in differences). Here, the propensity score matching multiple difference approach (PSM-DID) was employed to more precisely and effectively assess the reconstruction of county economic space by high-speed rail. The specific operations and models are as follows:

1.  To identify the group that corresponds with the county units that offer high-speed rail service, use PSM.
2.  The county units of the matched experimental group and control group were used for the DID model evaluation and analysis, and the following regression equation was obtained after combination:

$$\mathrm{Y}_{it}^{PSM} = \beta_0 + \beta_1 city_i + \delta_0 post_{it} + \delta_1 city_i * post_{it} + \beta_2 xlist_{it} + \mu_i + \lambda_t + \varepsilon_{it} \tag{1}$$

where $i$ denotes the region, $t$ represents the year, and $\mathrm{Y}_{it}$ is the dependent variable, indicating the economic development level of county $i$ in the year $t$ (including the agglomeration level measurement of the industrial enterprises and the secondary labor force's agglomeration degree). $city_i$ is an individual dummy variable of the county. If the policy of the high-speed rail opening affects county $i$, its value is 1, while the county not affected by the policy is 0. $post_{it}$ represents the time dummy variable in the processing period of the high-speed railway policy; the value of the county is 1 after the high-speed railway is opened and 0 if it is not. The interaction item ($city_i$ * $post_{it}$) represents the dummy variable of the county unit after the high-speed railway opening. Its coefficient $\delta_1$ is the difference between the impact of the high-speed railway opening on the treatment group and the control group, which this paper focuses on. $xlist_{it}$ is a group of control variables affecting the county unit economic development, and $\mu_i$ represents the individual fixed effect and is used to control the heterogeneity of the county units, while $\lambda_t$ represents the time fixed effect and is used to control the corresponding year of county units, and $\varepsilon_{it}$ is the residual phase.

### 3.3. Variable Selection

There are two types of indicators for county unit economic growth: multiple indicators and comprehensive single indicators. The study examines high-speed rail's effects on the urban unit economic development using the indicators used by Ahlfeldt and Feddersen [38], Redding and Tumer [39], and Kim [35]. It also confirms the above hypothesis regarding the patterns of enterprises and labor force agglomeration in county economic development. The multi-index analysis method was employed in assessment. We focused on how high-speed rail affects industrial firm dispersion as well as labor force movement and concentration in secondary sectors such as manufacturing. Several academics both domestically and internationally utilize the percentage of the total industrial output value to gauge the degree of regional industrial agglomeration when studying the production and agglomeration of county enterprises. For instance, to determine the industrial agglomeration level, Wen (2004) [40] and Yu Jin (2006) [41] both used the percentage of each region's industrial production value in the overall industrial GDP of the year as a variable. Since the data on the country's gross industrial output value were only available through 2012, the paper substituted the GDP for the corresponding year. Implementing the index was comparatively reasonable and dependable because the tangible and quantifiable aspects of manufacturing items have been perfected in statistical caliber and procedures. The proportion of workers in secondary industry units across the entire nation at the end of the year was used to

gauge the industrial labor force agglomeration degree in terms of the size, distribution, and concentration of the county labor force.

The DID model was employed to address some of the study object's endogeneity issues. However, to more precisely analyze the alterations and variations of the industrial enterprise and secondary labor agglomeration levels across counties with high-speed and non-high-speed rail, we cite the study by Shao et al. [6] on service sector agglomeration in the Yangtze River Delta region of China, the study by Dai and Hatoko [42] on the economic disparities between Switzerland and Japan regarding high-speed rail, and the study by Wang et al. [43] on high-speed rail service's effects on population flow and urbanization regarding industry and population by using control variables to fix the model's endogenous and sequential issues. In addition, the control variables were treated logarithmically to eliminate the collinearity issue. The final selection of control variables included the county's total population at the end of the year (lnpop), the gross regional product (lngdp), the national market potential (lnpot), the local market potential (lnlopot), and the completed amount of urban fixed assets investment (lnfid). To govern county economic development, industrial enterprise agglomeration, and secondary industry labor agglomeration, these factors were used as variables. For a description of the chosen variables and their selection, see (Table 1).

**Table 1.** Variable description.

| Classification | Variable | Symbol | Unit | Definition |
|---|---|---|---|---|
| Dependent variable | Agglomeration level of industrial enterprises | sec | % | Local industrial output value above designated size/annual GDP |
| | Concentration degree of secondary labor force | emp | % | Local employees in secondary industry units at the end of the year/national employees in secondary industry units at the end of the year |
| Independent variable | County unit dummy variable | $city_i$ | / | By 2019, the value of county units with high-speed rail service will be 1, otherwise 0 |
| | Policy processing period dummy variable | $post_{it}$ | / | From 2003 to 2019, the county value was 1 after the high-speed rail service and 0 before the high-speed rail service |
| | Interaction item | $city_i * post_{it}$ | / | After the high-speed rail line opens, the virtual variable in the county area is 1 in terms of time dimension; otherwise, it is 0. The virtual variable of the county area opened by high-speed rail is 1, else it is 0 in terms of region dimension |
| Control variable | Total population of the county at the end of the year | lnpop | Ten thousand people | County population at the end of the year, logarithm |
| | Gross regional domestic product | lngdp | Ten thousand CNY | Gross county product of corresponding year, logarithm |
| | National market potential | lnpot | / | The logarithm of the total retail sales of consumer goods in the local county divided by the sum of distances from other urban units (urban area, county area) to the local county unit ($mp_i = \sum_{j=1}^{R} RET_j d_{ij}^{-1}$, where $mp_i$ is the market potential of county $i$, $RET$ represents the total retail sales of social consumer goods in county, and $d_{ij}$ is the distance between county $i$ and county $j$ |
| | Local market potential | lnlopot | / | The logarithm of the total retail sales of consumer goods in a county divided by the distance to the nearest prefecture-level city ($lomp_i = RET_i/Nd_{ij}$, where $lomp_i$ is the local market potential of county i, RET represents the total retail sales of consumer goods in county, and $Nd_{ij}$ is the distance between county i and the nearest prefectural-level city j |
| | Total investment in urban fixed assets completed | lnfid | / | The amount of urban fixed assets investment completed in the corresponding year is logarithm |

Notes: sec, lnpop, lngdp, and lnfid all use 10,000 as the unit of measurement in variable calculation; since the number of secondary industry employees in the county is relatively small, the emp variable uses each person as the unit of measurement; in lnpot and lnlopot, the total retail sales of social consumer goods are measured in 10,000, and the distance is measured in kilometers.

*3.4. Description of Research Objects and Data*

3.4.1. Research Object and Scope

The essay primarily investigates how high-speed rail's launch has affected local economic growth (focusing on the interaction between industrial production and the corresponding secondary labor force). Thousands of county-level administrative divisions in China are becoming increasingly crucial with the declining market (Source: National Development and Reform Commission, China, https://www.ndrc.gov.cn/ (accessed on 1 July 2022)). A city's formation and growth cannot progress without considering administrative power. In mainland China, the term "urban establishment" refers to the administrative establishment system, which includes cities, counties, and municipalities directly under the central government. As of March 2020, there were 333 prefecture-level administrative divisions in China (excluding Hong Kong, Macao, and Taiwan) (Source: http://data.acmr.com.cn/member/city/city_md.asp (accessed on 28 July 2022)). These 300-plus cities have received the most attention in studies on China's high-speed rail network's effects [44–49]. We consider that county-level administrative divisions have evolved into the initial transfer point for non-agricultural enterprises and population agglomeration, because of China's huge area, enormous population, and the high-speed rail network constantly spreading and encompassing counties. Researchers studying high-speed rail should note China's thousands of county-level administrative units, which are crucial to the country's development. Geographically, economically, and in terms of management authority, the municipal district is closer to the central city than the county-level cities and counties under its jurisdiction; we list the primary urban units examined in this article as the county-level cities and counties governed by the central cities. Based upon this, the nation's cities are split into four categories of urban units, including high-speed urban area, non-high-speed urban area, high-speed county, and non-high-speed county, according to the administrative zoning size. The four categories of urban units are explained in detail below, and the geographic spatial distribution of the four types of urban units during the research year (up to 2019) is visualized in Arcgis10.8 (Figure 4) (Standard map base source: http://bzdt.ch.mnr.gov.cn/ (accessed on 30 July 2022)). To investigate the economic growth trajectory and population distribution rule of high-speed rail counties under the influence of high-speed rail, we selected 1791 county-level urban units (county-level cities and counties) across the country as empirical research subjects. The four types of urban units were classified as follows:

1.  High-speed urban area: If any of the sub-provincial-level cities and prefecture-level city districts opened high-speed railways, the whole prefecture-level city district was divided into high-speed railway central city units, such as Beijing, Shanghai, Tianjin, and Chongqing.
2.  Non-high-speed railway urban areas: In contrast to the high-speed urban areas, the prefecture-level city municipal districts were divided into non-high-speed railway urban units if there was no high-speed service in any section of the sub-provincial city or prefecture-level city municipal district.
3.  High-speed county: In addition to the central city units mentioned above, if county-level cities, county-level administrative units, and the units below had high-speed rail service, they were classified as high-speed rail counties.
4.  Non-high-speed county: Conversely, if county-level cities, county-level administrative units, and the units below had no high-speed rail service, they were classified as non-high-speed counties.

3.4.2. Data Sources

The panel data of county area city units, constituted of 1791 counties and county-level cities under the jurisdiction of the sample urban areas in China from 2003 to 2019, were used as the county data in this article. The administrative region zoning by the end of 2020 was used as the benchmark, considering China's administrative divisions. Regarding high-speed rail, information about China's high-speed rail projects from 2003 to 2019 was

gathered and organized using the websites for "Train Schedule 2010," "Train Schedule 2020," the National Railway Administration of China, and the High-Speed Railway Network. For details on data gathering, see the table below (Table 2).

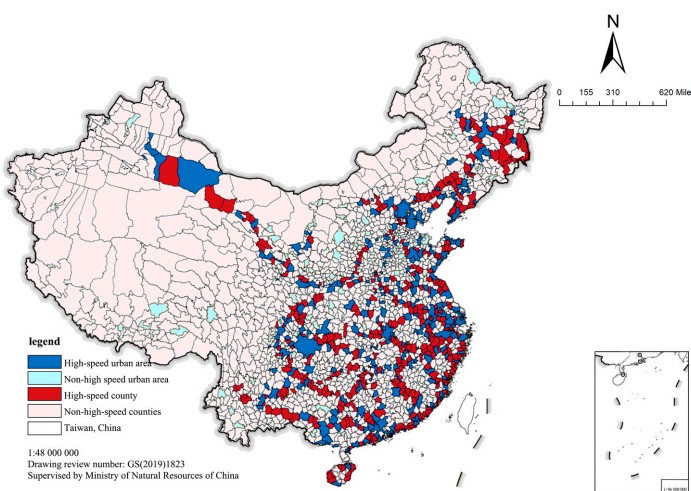

**Figure 4.** Geographical spatial distribution of four types of urban units (by the end of 2019). Note: This map is based on the standard map No. GS (2019)1823 downloaded from the Standard Map Service website of the Department of Natural Resources. The base map is unchanged.

**Table 2.** Summary table of datasets involved in the study.

| Data Set | Data Sources | Year |
|---|---|---|
| China County Statistical Yearbook (County and City Volume) | Summary by Office for National Statistics: www.stats.gov.cn (accessed on 1 July 2022). | 2003–2019 |
| Base map of county-level administrative divisions in China | Chinese government website: www.gov.cn(accessed on 19 July 2022) | By 2020 |
| County city unit base map | Based on county-level administrative division file arrangement | By 2020 |
| City centers of prefecture-level cities and counties | Summary by Chinese government website: www.gov.cn (accessed on 17 July 2022). | By 2020 |
| The distance between counties and urban units | Calculation based on Arcgis10.8 software | By 2020 |
| Timetable of high-speed rail | Summary by China Railway Network: www.12306.cn (accessed on 17 July 2022). | 2010, 2020 |
| Data of the opening of high-speed rail lines over the years | Summary by National Railway Administration: www.nra.gov.cn; High-speed rail network: www.gaotie.cn (accessed on 14 July 2022) | 2003–2019 |
| Data of the opening of high-speed rail stations over the years | Summary by National Railway Administration: www.nra.gov.cn; High-speed rail network: www.gaotie.cn (accessed on 14 July 2022) | 2003–2019 |

### 3.4.3. Sample Matching

We performed PSM matching between the treatment and control groups prior to the empirical analysis. Consequently, 358 county units that had opened high-speed rail between 2003 and 2019 constituted the treatment group, while 1433 county units that had not opened high-speed rail during that time constituted the control group. The probit model was used to estimate the P-score. The weight was calculated using the nearest-neighbor tendency matching approach, and the "on support" condition was appended (see Appendix A (Tables A1–A3 and Figure A1) for specific PSM results).

The kernel density function's distribution curve before and after the nearest-neighbor tendency was drawn (see Figure 5). According to Figure 5a in the comparison chart, the distribution of counties in the treatment group was extremely loose before PSM matching was carried out. In contrast, the control group's P-score kernel density distribution was

clearly concentrated and skewed to the left. The impact of high-speed rail on the P-score probability density distribution in counties differed widely. The results of the multi-stage DID analysis would unavoidably have been seriously affected by sample selection bias if PSM matching were not used and the concentration level of the industrial enterprises and the secondary industry labor force between the two groups of the county samples were directly compared. The findings' reliability would thus be impacted. We compared the characteristics of the two county sample groups in five dimensions: GDP, market potential, distance from the county center to the closest prefecture-level city, and the amount of urban fixed assets investment that was completed at the end of the year. Furthermore, we employed the nearest-neighbor method. After matching, the retained samples' features were identical in every way, and the selectivity bias was essentially removed (see Figure 5b).

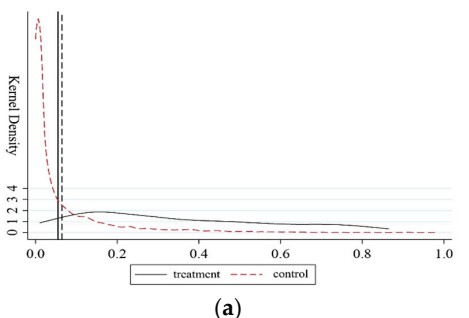
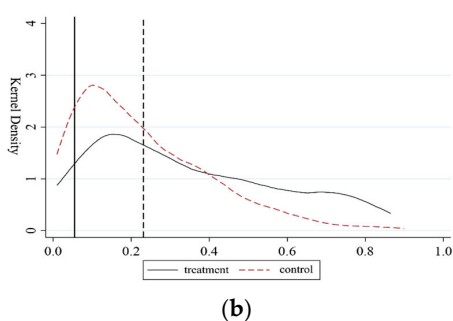

(**a**)                                      (**b**)

**Figure 5.** Comparison of kernel density distribution of propensity scores between the treatment group and the control group before and after nearest-neighbor propensity matching. (**a**) Kernel density of propensity score before matching. (**b**) Kernel density of propensity score after matching).

## 4. Empirical Analysis

### 4.1. Impact Analysis of High-Speed Rail on County Economic Development Based on PSM-DID

The section focuses on the heterogeneity of high-speed railway construction's influence on the labor force agglomeration of industrial enterprises and secondary industries. To test the PSM sample matching's effectiveness, a mix of OLS baseline regression 1 (see model 1), OLS baseline regression after PSM matching (see model 2), fixed-effect model baseline regression (see model 3), DID regression using samples satisfying the common support, which is the tendency of scores of the treatment group and the control group to have a large common range of values (see model 4), multi-phase PSM-DID regression, and samples satisfying the common supporting hypothesis (see model 5) were used. After PSM-DID estimation, the standard error mean of the control variables was reduced, and the interaction term's regression coefficient was larger than that of the ordinary DID estimated by the fixed-effect model's regression. Overall, we think that PSM-DID's estimation effect was better.

#### 4.1.1. Impact of High-Speed Rail on Industrial Agglomeration in Counties

The section focuses on the industrial agglomeration differences between counties served by and those not served by high-speed rail (see Table 3). The following outcomes were found: (1) With the exception of model 1, all five models' regression coefficients for the interaction terms of the explanatory variables (city*post) were significant above the 10% level, showing that the regression results were still reliable even after accounting for the selectivity bias issue. The interaction terms of the matching results of PSM-DID were also significant at the 1% level. This demonstrates how individual variances between counties can skew the research findings. (2) Qin's research (2017) also points out that high-speed railway construction reshapes counties' economic activities. Moreover, it has certain negative effects on county economic development [50]. (3) The regression results of the control variables show that the gross regional product, the size of the national market, and the completion of urban fixed asset investments have a significant impact on the industrial

agglomeration in counties, whereas the total population at the end of the year and the size of the local market have little effect. The possible reason is that a county's industrial development is not only related to the nearest central city, but also closely related to its own level of economic development, urban construction, and market potential in a larger scope.

4.1.2. High-Speed Rail's Influence on the Agglomeration of Secondary Industry Labor Force in Counties

The section focuses on the differences in secondary labor agglomeration between counties with and without high-speed rail because of high-speed rail installation (see Table 4). The following are the outcomes: (1) The regression coefficient of model 5 shows obvious significance at the 5% level, indicating that high-speed rail opening has a significant impact on counties' secondary industry labor force agglomeration. Meanwhile, the regression coefficient is positive; indicating that, in contrast to the result of the industrial enterprises, high-speed railway opening promotes the concentration of secondary labor force in the county. The possible reason is that the county area attracts the secondary labor force of the central city and the surrounding agricultural population. (2) Among the control factors, the size of the fixed asset investment in metropolitan areas, the size of the national market potential, and the size of the local market potential all significantly affect the labor force distribution in the secondary industry. The secondary industrial labor force concentration in the county is also not significantly impacted by the total population at the end of the year or the gross regional product. In contrast, it is clear from the coefficient that the local market potential has a very beneficial effect on the secondary industry labor force concentration in the county when compared to the degree of industrial agglomeration.

**Table 3.** Evaluation results of impact of high-speed rail opening on industrial agglomeration in county area based on multi-phase PSM-DID model.

|  | Model 1 | Model 2 | Model 3 | Model 4 | Model 5 |
|---|---|---|---|---|---|
|  | ols1 | ols2 | fe | Common Support | Psm-Did |
| citypost | −0.00013 | −0.00015 *** | −0.00005 * | −0.00005 * | −0.00007 *** |
|  | (−1.52423) | (−4.12649) | (−1.74838) | (−1.74838) | (−3.21568) |
| lnpop | −0.00011 *** | −0.00007 * | −0.00022 *** | −0.00022 *** | −0.00008 |
|  | (−3.96557) | (−1.81789) | (−4.33313) | (−4.33313) | (−1.20055) |
| lngdp | 0.00036 *** | 0.00115 *** | 0.00020 *** | 0.00020 *** | 0.00072 *** |
|  | (6.20586) | (21.13444) | (6.38093) | (6.38093) | (9.97349) |
| lnpot | 0.00090 *** | 0.00010 | −0.00026 *** | −0.00026 *** | −0.00049 *** |
|  | (3.58495) | (1.07191) | (−3.14396) | (−3.14396) | ($-1.0 \times 10^2$) |
| lnlopot | −0.00002 | 0.00002 | −0.00003 | −0.00003 | 0.00005 |
|  | (−0.89649) | (0.91598) | (−1.26638) | (−1.26638) | (0.61204) |
| lnfid | −0.00015 *** | −0.00054 *** | 0.00000 | 0.00000 | −0.00006 *** |
|  | (−5.44759) | (−1.8 e + 01) | (0.04726) | (0.04726) | (−3.01223) |
| N | 3706 | 833 | 3706 | 3706 | 833 |
| Adj. $R^2$ | 0.50289 | 0.57250 | 0.27987 | 0.27987 | 0.37081 |

Note: Numbers in brackets are standard error. *** and * are significant at the level of 1, and 10%, respectively.

**Table 4.** Evaluation results of impact of high-speed rail opening on secondary industry labor force agglomeration in county based on multi-phase PSM-DID model.

|  | Model 1 | Model 2 | Model 3 | Model 4 | Model 5 |
|---|---|---|---|---|---|
|  | ols1 | ols2 | fe | Common Support | Psm-Did |
| citypost | −0.00006 *** | −0.00004 * | 0.00002 | 0.00002 | 0.00004 ** |
|  | (−3.51918) | (−1.78358) | (1.01943) | (1.01943) | (2.28175) |
| lnpop | 0.00010 *** | 0.00021 *** | −0.00001 | −0.00001 | 0.00008 |
|  | (12.04140) | (8.70324) | (−0.29664) | (−0.29664) | (1.59268) |

**Table 4.** *Cont.*

|  | Model 1 | Model 2 | Model 3 | Model 4 | Model 5 |
|---|---|---|---|---|---|
|  | ols1 | ols2 | fe | Common Support | Psm-Did |
| lngdp | 0.00016 *** | 0.00063 *** | 0.00004 ** | 0.00004 ** | 0.00008 |
|  | (14.78234) | (17.36344) | (2.40126) | (2.40126) | (1.33910) |
| lnpot | 0.00108 *** | 0.00060 *** | −0.00005 | −0.00005 | −0.00023 *** |
|  | (41.40345) | (9.34073) | (−0.69452) | (−0.69452) | (−6.28128) |
| lnlopot | −0.00001 ** | −0.00002 | 0.00002 | 0.00002 | 0.00012 ** |
|  | (−2.26344) | (−1.13555) | (1.02721) | (1.02721) | (2.00929) |
| lnfid | −0.00010 *** | −0.00036 *** | 0.00001* | 0.00001 * | −0.00003 * |
|  | $(-1.8 \times 10^2)$ | $(-1.8 \times 10^2)$ | (1.90752) | (1.90752) | (−1.82140) |
| N | 3706 | 833 | 3706 | 3706 | 833 |
| Adj. $R^2$ | 0.66672 | 0.68823 | 0.02683 | 0.02683 | −0.09127 |

Note: Numbers in brackets are standard error. ***, ** and * are significant at the level of 1, 5, and 10%, respectively.

### 4.2. Analysis of the Time-Delay Impact of High-Speed Rail on County Economic Development

According to the above empirical analysis, high-speed railway opening has a long-term impact on county economic development. Therefore, in this paper, by referring to the event study method proposed by Clarke and Schythe [51], the delay impact of high-speed railway on the county industrial agglomeration level and secondary labor agglomeration degree is explored.

#### 4.2.1. Analysis of the Delay Effect of High-Speed Railway on Industrial Agglomeration in County

High-speed rail opening has a negative impact on the industrial agglomeration level (sec) in counties to a certain extent. However, whether the negative impact has a time lag is verified in this part (see Figure 6). The concrete results show that: (1) From the significant regression coefficient level, high-speed rail opening has a leading effect on and a role in promoting industrial enterprise development. At the same time, high-speed rail opening has obvious delay effect on the county industry's agglomeration and development. When the lag period is the second phase (i.e., the second year), a county's industrial agglomeration level show obvious correlation with the high-speed rail opening, thus indicating that a county's high-speed rail opening will have a long-term negative impact on the industrial agglomeration. (2) From the interaction term's regression coefficient, the leading period is the promoting effect, while the lagging period is the negative effect. The possible reason is that Chinese counties' development levels are relatively behind those of others at present, and that high-speed rail opening has intensified the siphon effect of central cities.

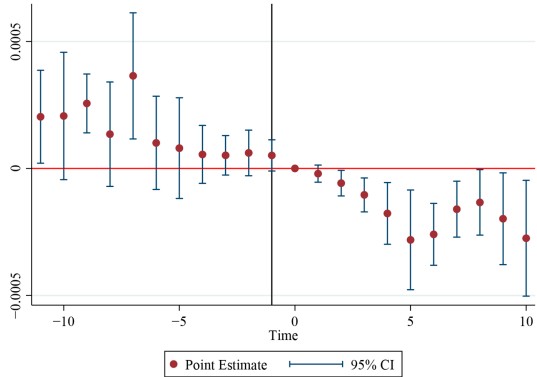

**Figure 6.** Regression results of the delayed impact of high-speed railway opening on industrial agglomeration in counties. Note: Since there was only one high-speed train in 2003 to 2008, the lag time was only 10 years in 2019 minus 2008.

#### 4.2.2. Analysis of High-Speed Railway's Time-Delay Impact on Counties' Secondary Industry Labor Force Agglomeration

High-speed railway opening promotes secondary labor force agglomeration (emp) to high-speed railway counties. This part continues to explore the delay of high-speed railway to secondary labor force agglomeration in counties and verifies its dynamic lag (see Figure 7). The results show that: In Section 4.1.2, which examines the impact of secondary labor force agglomeration in county units in the year of high-speed railway opening, we discovered that the introduction of high-speed railway encouraged secondary labor force agglomeration. However, we find that it is not the case from the dynamic time-delay analysis of the secondary labor force agglomeration. The influence of high-speed railway on the agglomeration of secondary labor force in county units reveals an erratic state. The elimination of China's demographic dividend and the restriction of the household registration system in the panel data analysis from 2003 to 2019 may be the cause, which diminished the influence of high-speed rail on secondary labor flow.

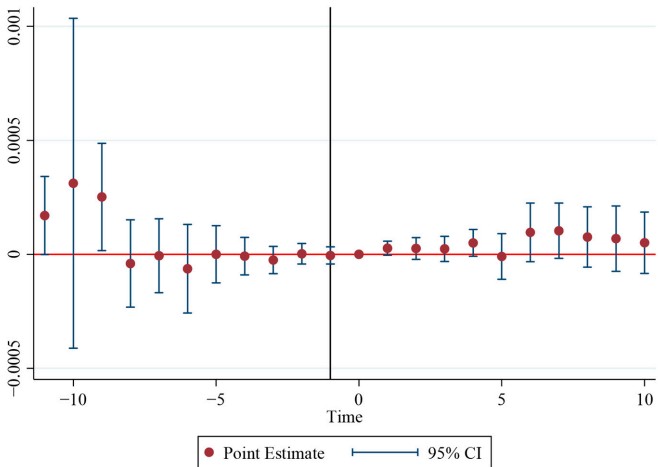

**Figure 7.** Regression results of the delayed impact of high-speed railway opening on secondary industry labor force agglomeration in counties. Note: Since there was only one high-speed train in 2003 to 2008, the lag time was only 10 years in 2019 minus 2008.

### 5. Robustness Test

*5.1. Endogenic Processing*

Excluding endogenous problems' influence is a prerequisite for DID analysis. We mainly referred to Huber and Steinmayr [52] and Jun Zhang [53] by investigating counties with less discourse power and higher randomness as research subjects; two major measures of PSM matching analysis were carried out on the experimental and control groups of the county units to ensure the selection of the county units met the three measures of the randomness hypothesis test, processing the matching of other relevant characteristics of samples, and carrying out a collinearity test of the variables to cause the experimental results to be more reliable. The specific measures are as follows:

1.  The inauguration of high-speed rail must be a quasi-natural experiment to pass the random hypothesis test. This is the basis of applying the multi-phase DID model analysis. In his article on high-speed rail's effect on county economic development, Jun Zhang noted the close connection between the planning and construction of high-speed rail lines and stations and the degree of regional economic development, as well as their geographic location and other factors. The likelihood that a central city will open high-speed rail increases with its economic power and political clout. Given this, we opted not to use the prefecture-level cities that many academics have chosen to analyze to examine high-speed rail's effects on urban economic development. To address the endogenous issue, we instead selected counties with lower discourse power and higher randomness as the research subjects.

2. To further limit endogeneity, a multi-phase PSM-DID model based on county unit matching was created in this study. The population, level of economic development, and market potential in county units vary greatly, and this unit heterogeneity causes a clear bias in policy effect estimation. Up to a point, the sample selectivity bias could be reduced by matching the propensity scores of the experimental and control groups based on the characteristics of county units.

3. To test whether there was collinearity among the variables, the collinearity diagnosis was conducted (Table 5). Except for the gross regional product, the VIF value was no more than 6, which proved that there was no collinearity among the variables or that there was to a lesser extent.

**Table 5.** Collinearity diagnostics table.

|  | Model 1 | Tolerance | VIF | Model 2 | Tolerance | VIF |
|---|---|---|---|---|---|---|
| dependent variable | sec |  |  | emp |  |  |
|  | emp | 0.325 | 3.079 | sec | 0.422 | 2.368 |
|  | lnpop | 0.326 | 3.066 | lnpop | 0.318 | 3.145 |
| independent variable | lngdp | 0.094 | 10.687 | lngdp | 0.085 | 11.832 |
|  | lnpot | 0.318 | 3.144 | lnpot | 0.382 | 2.621 |
|  | lnlopot | 0.173 | 5.786 | lnlopot | 0.173 | 5.788 |
|  | lnfid | 0.252 | 3.968 | lnfid | 0.249 | 4.021 |

### *5.2. Parallel Trend Test*

The DID model's basic assumption is that the treatment and the control groups have parallel trends; that is, the two county unit groups should have the same trend before the high-speed railway opening. To test the robustness of high-speed rail opening's influence, this paper adopts the method of changing the window width before and after the county high-speed rail opening for verification. The specific formula was as follows:

$$Y_{it} = \beta_0 + \delta_1 \sum_{k \geq 3}^{2} city_i * post_{i,tc0+k} + \beta_2 xlist_{it} + \mu_i + \lambda_t + \varepsilon_{it} \tag{2}$$

where $tc0$ represents the year of the high-speed railway opening, and $t - tc0 = k = -3$, $-2$, $-1$, 0, 1, 2, 2, 3 represent the county units set dummy variables for three years, two years, one year, one year, two years, and three years before the high-speed railway opening, and construct the interaction terms between the corresponding county units and the time dummy variables. Parallel trend tests were carried out on the industrial agglomeration level (sec) and the secondary labor agglomeration degree (emp) of the county units, respectively (see Table 6 and Figure 8). The results show that: ① From the regression coefficient, the regression coefficients of the industrial agglomeration level and secondary labor agglomeration degree are not significant in the three years before the high-speed railway opening, while the baseline group (to completely exclude the collinearity problem, the first period before the policy is usually selected as the baseline group) is significant in the later years. The results indicated that the county units with and without high-speed rail services had the same time trend at least three years before the high-speed rail service. The parallel trend test results verified the multi-period PSM-DID regression results' robustness. ② The parallel trend test chart shows the dynamic economic effects between different years under the high-speed railway policy's impact. The chart indicates that the interaction term's coefficients were not significantly different from 0 before the high-speed railway opening, and that the confidence intervals all contain 0 values, indicating that there was no significant difference between the county units of the experimental and control groups before the high-speed railway opening, which satisfies the hypothesis of the parallel trend. In addition, the parallel test trends of sec and emp after implementing the policy were opposite, thus indicating that the high-speed rail opening significantly differed in its impact on the industrial development and secondary labor force agglomeration across counties.

**Table 6.** Parallel trend test table of industrial agglomeration level and secondary labor agglomeration degree in county units.

| | (1) | (2) |
|---|---|---|
| Variables | Sec | Emp |
| pre_3 | $1.59 \div 10^5$ | $-1.41 \div 10^5$ |
| | $(3.57 \div 10^5)$ | $(3.32 \div 10^5)$ |
| pre_2 | $-3.47 \div 10^5$ | $-1.38 \div 10^5$ |
| | $(3.00 \div 10^5)$ | $(1.85 \div 10^5)$ |
| current | $-3.23 \div 10^5$ | $1.11 \div 10^5$ |
| | $(2.64 \div 10^5)$ | $(1.80 \div 10^5)$ |
| post_1 | $-4.69 \div 10^5$ | $3.53 \div 10^5$ |
| | $(3.69 \div 10^5)$ | $(2.33 \div 10^5)$ |
| post_2 | $-8.24 \div 10^5$ ** | $3.30 \div 10^5$ |
| | $(4.15 \div 10^5)$ | $(3.07 \div 10^5)$ |
| post_3 | $-0.000112$ ** | $4.09 \div 10^5$ |
| | $(4.42 \div 10^5)$ | $(3.35 \div 10^5)$ |
| post_4 | $-0.000168$ ** | $6.99 \div 10^5$ ** |
| | $(7.73 \div 10^5)$ | $(3.41 \div 10^5)$ |
| post_5 | $-0.000174$ ** | $9.92 \div 10^5$ * |
| | $(7.19 \div 10^5)$ | $(5.61 \div 10^5)$ |
| lnpop | $-2.95 \div 10^5$ | $5.54 \div 10^5$ |
| | $(7.44 \div 10^5)$ | $(7.04 \div 10^5)$ |
| lngdp | $0.000697$ *** | $8.59 \div 10^5$ |
| | $(0.000127)$ | $(6.40 \div 10^5)$ |
| lnpot | $-0.000467$ *** | $-0.000251$ ** |
| | $(0.000152)$ | $(0.000107)$ |
| lnlopot | $6.47 \div 10^5$ | $0.000123$ |
| | $(0.000163)$ | $(0.000117)$ |
| lnfid | $-4.97 \div 10^5$ ** | $-3.12 \div 10^5$ |
| | $(2.12 \div 10^5)$ | $(1.91 \div 10^5)$ |
| Constant | $-0.00914$ *** | $-0.00146$ |
| | $(0.00161)$ | $(0.00128)$ |
| Observations | 810 | 810 |
| R-squared | 0.968 | 0.967 |

Note: Numbers in brackets are standard errors. ***, ** and * are significant at the level of 1, 5, and 10%, respectively.

*5.3. Placebo Test*

There were individual differences over time between the county units, regardless of whether high-speed rail was opened, which led to systematic bias in the regression results. The discrepancies in the industrial concentration level (sec) and secondary labor concentration degree (emp) between the two groups of county units should not change over time if the high-speed rail development did not occur. The discrepancies in the industrial concentration level (sec) and secondary labor concentration degree (emp) between the two county unit groups should not change over time if high-speed rail development did not occur. The discrepancies in the industrial concentration level (sec) and secondary labor

concentration degree (emp) between the two groups of county units should not change over time if high-speed rail development did not occur. To verify the regression results' reliability, Liu Ruiming et al. (2020) employed a method that is extensively used in the placebo test to randomly produce group tests in the research [54]. The kernel density coefficient and *p*-value scatter plot of the virtual policy variables were compared and examined to see whether there were any significant differences between them and the actual values by randomly generating the processing groups of counties that opened high-speed rail and repeating the regression 1000 times (see Figure 9). As seen from the figure, both the industrial agglomeration level (sec) and the concentration degree of the secondary industry labor force (emp) indicate that, in the case of random sampling, the regression coefficients close to the real benchmark (solid line) tsec = −0.0000457 and temp = 0.0000199 were all small probability events, thus suggesting that high-speed rail opening affected the regional economy. The aforementioned empirical investigation results are reliable and the placebo test was passed.

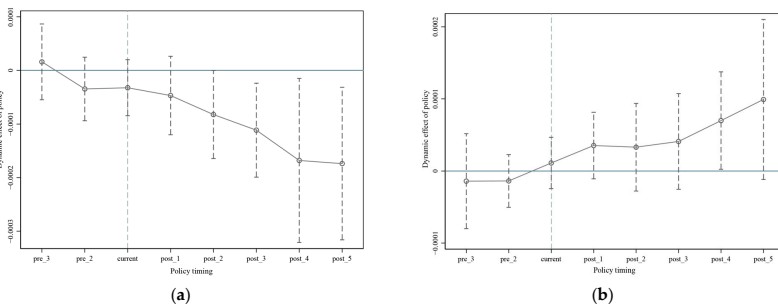

**Figure 8.** Parallel trend test diagram of industrial agglomeration level and secondary labor agglomeration degree in county units. (**a**) Test chart of parallel trend of industrial agglomeration level. (**b**) Test chart of parallel trend of concentration degree of secondary industry labor force.

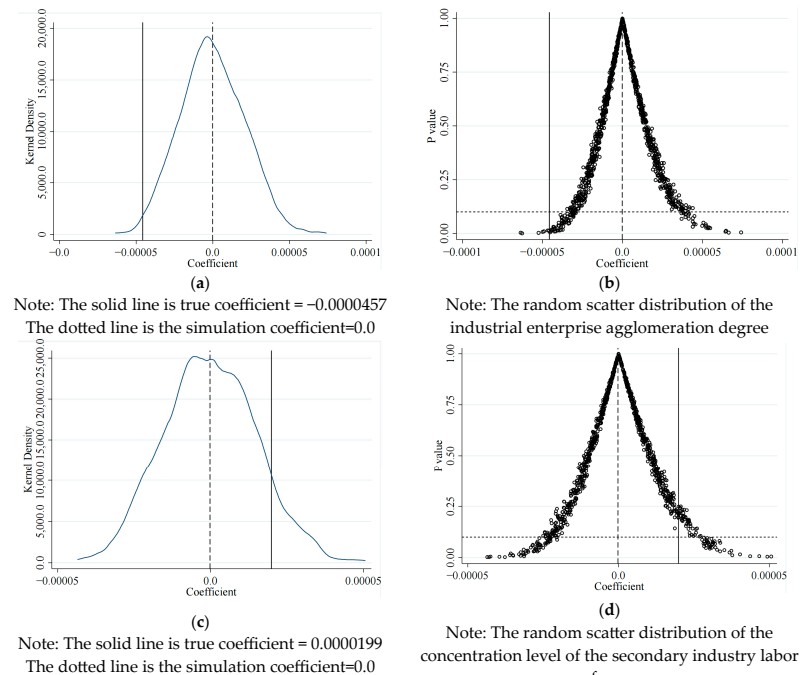

**Figure 9.** Placebo test chart of industrial agglomeration level and secondary labor agglomeration degree in county units. (**a**) Kernel density coefficient diagram of the industrial concentration level. (**b**) Scatter plot of *p*-value of placebo test for the industrial concentration level. (**c**) Kernel density coefficient diagram of the concentration degree of secondary labor force. (**d**) Scatter plot of *p*-value of placebo test for the concentration degree of secondary labor force.

## 6. Research Conclusions and Policy Implications

High-speed rail opening eases the historically severely imbalanced regional economy growth brought on by China's high flow cost of labor and other generating factors. The key to the balanced and high-quality development of China's economy in the future is the transformation and upgrading of county industrial and demographic structures. In the context of China's rapid urbanization and regional development transformation, the multi-mode transportation system, including airports, high-speed railways, rail transit, expressways, and buses, has developed into a crucial foundation for coping with the "congestion costs" of big cities, such as urban industrialization and rising housing prices. Many studies and statistical evidence have demonstrated how developing a high-speed rail network has aided in the trend toward suburbanization, regional integration, and cross-regional travel. For instance, Xiongbin Lin and Yuan Lu discovered, through a questionnaire study, that 7% of commuters use the Beijing–Tianjin cross-city high-speed railway [55]. This study examines the new route of county transformation and upgrading in our nation regarding high-speed rail's impact on the county industrial and secondary labor agglomeration. The study views the launch of high-speed rail as a sort of natural experiment. A multi-period PSM-DID model was built to empirically investigate the effects of introducing high-speed rail on the concentration of industrial firms and the secondary labor force in the counties using panel data from 1791 non-central urban units in China from 2003 to 2019.

The findings indicated that introducing high-speed rail changed counties' economic landscapes, had an adverse effect on industrial agglomeration, and had a favorable effect on the secondary labor concentration. According to the examination of high-speed rail's time delay, industrial firm development in county regions is normally negatively impacted by high-speed rail for a long period before turning around eight years later. High-speed rail has a long-term positive effect on the concentration of the secondary labor force, but does not show an obvious trend of orderly concentration. Here, we found China's high-speed rail counties are currently developing primarily into satellite cities formed by population agglomeration, or "sleeping cities". Developing a high-speed rail network has, according to pertinent studies, accelerated the national gradient transfer of low-end manufacturing businesses to less-developed regions [56–59]. The "siphon effect" of center cities prevents industrial units developing in neighboring cities; however, on the small scale of central cities and surrounding counties, the impact of intercity high-speed rail construction on neighboring county development is minimal. Additionally, we investigated how the potential of the national and local markets affected the county's economic growth. We discovered that, while the local market potential attracted a secondary labor force concentration, the national level mainly had a negative impact on the county's industrial development.

High-speed rail's effect on the county economic development was examined using a theoretical and an econometric model. Owing to the restricted research capacity and data collection, the research method still has major flaws that need to be addressed, notably the following two main points: (1) The theoretical framework and model need to be further refined. Cities are intricate systems. The variability of numerous significant economic determinants and unique county development features is still ignored by the theoretical framework of the interaction between county economic development and the high-speed rail network that the study created. (2) The data processing and characterization variable selection need to be further studied. Many county units with missing data were eliminated because of the small number of county samples and the lengthy period (2003–2019). The research results' accuracy differed from the real results due to a linear interpolation approach being employed to supplement a limited number of county units with missing data. The data availability also restricted the choice of variables. We propose that, in the future, research into the scope and interactions of multivariate factors, the data's veracity, and the use of high-speed rail for mass transit in the metropolitan region be performed in greater depth.

**Author Contributions:** Conceptualization, F.C. and H.H.; methodology, F.C. and H.H.; software, F.C.; validation, F.C.; formal analysis, F.C.; investigation, F.C.; resources, H.H.; data curation, F.C.; writing—original draft preparation, F.C.; writing—review and editing, F.C. and H.H.; visualization, F.C.; supervision, H.H.; project administration, H.H.; funding acquisition, H.H. All authors have read and agreed to the published version of the manuscript.

**Funding:** This research was funded by the Center for Balance Architecture of Zhejiang University (Project No: K Heng 20203512-02B, Index and planning methods of resilient cities).

**Data Availability Statement:** The data presented in this study are available upon request from the authors.

**Conflicts of Interest:** The authors declare no conflict of interest.

**Appendix A**

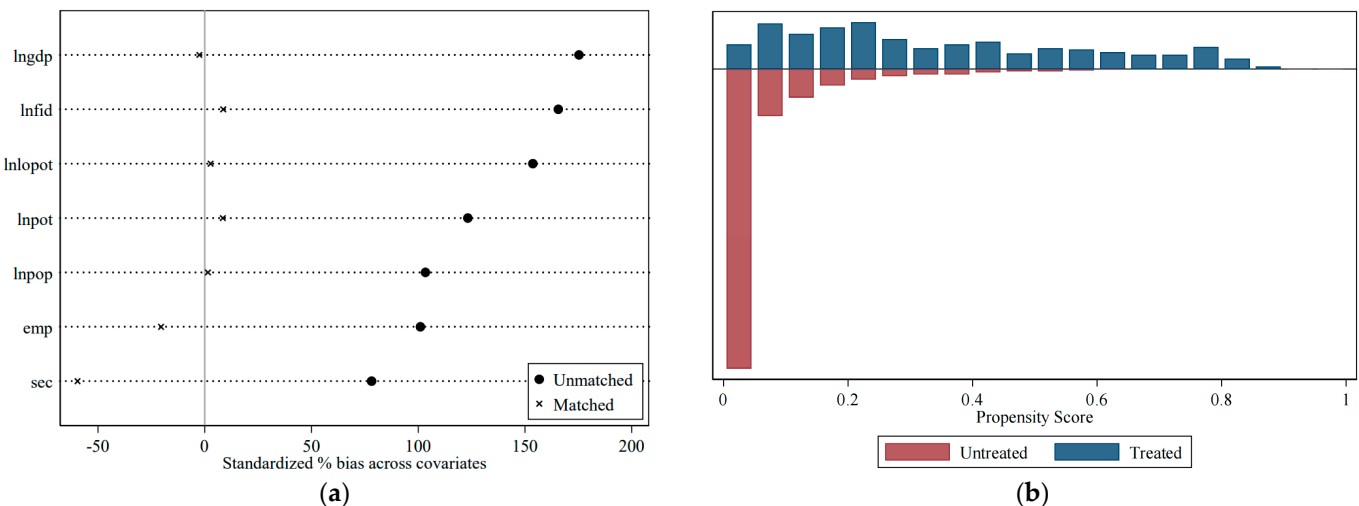

(**a**)　　　　　　　　　　　　　　　　　　　　(**b**)

**Figure A1.** PSM balancing table. (**a**) PSM equilibrium test diagram. (**b**) P-score balance test chart.

**Table A1.** PSM-nearest-neighbor matching result.

| Citypost | Coefficient | Std. Err. | z | *p* > |z| | [95% Conf. | [Interval] |
|---|---|---|---|---|---|---|
| lnpop | −0.0404 | 0.1414 | −0.2900 | 0.7750 | −0.3175 | 0.2367 |
| lngdp | 1.5066 | 0.2019 | 7.46000 | 0.0000 | 1.1108 | 1.9024 |
| lnpot | −0.1788 | 0.3579 | −0.5000 | 0.6170 | −0.8802 | 0.5226 |
| lnlopot | 0.0463 | 0.0976 | 0.4700 | 0.6350 | −0.14496 | 0.2376 |
| lnfid | 0.4051 | 0.1035 | 3.9100 | 0.0000 | 0.2022 | 0.6079 |
| _cons | −29.4528 | 2.3367 | −12.6000 | 0.0000 | −34.0327 | −24.8729 |

**Table A2.** PSM-nearest-neighbor matching result.

| Variable | Sample | Treated | Controls | Difference | S.E. | T-Stat |
|---|---|---|---|---|---|---|
| sec | Unmatched | 0.0007 | 0.0002 | 0.0005 | 0.0000 | 16.4200 |
| | ATT | 0.0007 | 0.0011 | −0.0004 | 0.0001 | −5.6500 |
| | ATU | 0.0003 | 0.0002 | −0.0001 | . | . |
| | ATE | . | . | −0.0001 | . | . |
| emp | Unmatched | 0.0008 | 0.0003 | 0.0005 | 0.0000 | 21.5500 |
| | ATT | 0.0008 | 0.0010 | −0.0001 | 0.0001 | −2.1300 |
| | ATU | 0.0003 | 0.0002 | −0.0001 | . | . |
| | ATE | . | . | −0.0001 | . | . |
| | Untreated | 3373 | 3373 | | | |
| | Treated | 333 | 333 | | | |
| | Total | 3706 | 3706 | | | |

Table A3. Descriptive statistics.

| Variable | Obs | Mean | Std. Dev. | Min | Max |
|---|---|---|---|---|---|
| id | 3706 | 109.5 | 62.939 | 1 | 218 |
| year | 3706 | 2011 | 4.9 | 2003 | 2019 |
| sec | 3706 | 0 | 0.001 | 0 | 0.004 |
| emp | 3706 | 0 | 0 | 0 | 0.003 |
| hsr | 3706 | 498.839 | 869.451 | 0 | 2019 |
| citypost | 3706 | 0.09 | 0.286 | 0 | 1 |
| lnpop | 3706 | 3.469 | 0.913 | 0.693 | 5.094 |
| lngdp | 3706 | 13.318 | 1.305 | 8.971 | 16.759 |
| lnpot | 3706 | 0.162 | 0.25 | 0 | 3.321 |
| lnlpot | 3706 | 8.228 | 1.827 | 2.499 | 14.463 |
| lnfid | 3706 | 12.678 | 1.568 | 4.159 | 28.597 |
| tmp | 3706 | 0.503 | 0.287 | 0 | 1 |
| pscore | 3706 | 0.09 | 0.154 | 0 | 0.984 |
| treated | 3706 | 0.09 | 0.286 | 0 | 1 |
| support | 3706 | 1 | 0 | 1 | 1 |
| weight | 833 | 4.449 | 32.598 | 0.333 | 538.667 |

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
