# Peer review of "The Impact of High-Speed Rail on Economic Development: A County-Level Analysis"

_land, doi:10.3390/land12040874_

Round 1

Reviewer 1 Report

This study tries to explore the impact of high-speed rail on the agglomeration of industrial enterprises and the secondary labor force in the 1791 county units in China. The authors employed a PSM-DID model based on the panel data from 2003 to 2019.

The study is of interest to international readers, as China has developed its high-speed rail infrastructure far more than any other country in the world. However, I think the following issues should be addressed before publication:

1. First of all, I think proofreading is needed to make the language of the article more fluent and to correct sentence constructions and grammatical errors.

2. The literature review section should be enriched with examples from different parts of the world. The differences and similarities between the experiences of different countries and the experience of China should be revealed.

3. Research questions and hypotheses in the analysis framework should be supported by findings from the literature and statistical data.

4. The authors often mention the “Siphon effect” and “trickle effect” in the article. What this means in the context of the article should be detailed in a few sentences.

5. Information on the use of high-speed trains in China should be shared. Are high-speed trains used as a commuting mode in China? Are there data/statistics on those using the high-speed train for their daily commute? What is the share of high-speed trains in total freight and passenger transportation? How do high-speed train lines in the country differ in terms of capacity and usage? The answers to these questions will also contribute to the interpretation of the study findings.

6. The authors use two different units of analysis in the study (County and urban area) What is the difference between county and urban area in China? What is the effect of including two different analysis units in the analysis simultaneously on the results?

7. Information about each data set used in the study should be shared in a table (Data source, years of data collection, etc.)

8. While matching samples by the PSM method, authors say that they use some criteria such as population, gross product, market potential and other characteristics. What are other characteristics they used in matching samples? For example, did they use the distance of counties to the biggest economic hubs in China?

9. Evaluating all districts that do not have a high-speed train station in the same category may lead to erroneous evaluations. Instead, a variable showing the distance of the county center to the nearest high-speed train stop, the nearest urban area or the nearest metropolitan area can be added to the analysis.

10. In the conclusion and discussion part, the limitations of the study and suggestions for future studies should be included.

Author Response

We would like to thank you for your thoughtful review, understanding of the language and expression being underdeveloped, recognition of the research. The valuable and constructive feedback that addressed several research limitations, which has significantly improved the presentation, the clarity, coherence, and persuasiveness of our manuscript. During the process of revising our work, we carefully reconsidered our language, theoretical concepts and frameworks, variables, as well as the limitations in our research conclusions. The explanation of what we have changed in response to the suggestions is given point by point in the following pages, please see the attachment.

Reviewer 2 Report

The paper assumes reader is in China or is intimately familiar with the Chinese urban and economics context. I don't think that's the case for the typical LAND reader. The last 2 paragraphs of the Introduction for example use jargon like "trickle down effect" of HSR and "siphon effect" of HSR without explaining. Similarly, "county-level cities and counties" is not clear to the international reader. 

The paper makes grand statements including words like "obviously" and "most" in areas that are at best underexplained, undertheorized, underhypothesized, or worse just not fully thought out. Why is HSR the MOST complex factor in location decisions? Why is the "siphon" effect OBVIOUS when you're positing a "trickle down effect"?

How about Path Dependence, especially for established agglomerations within a particular industry or cluster? There's evidence that moving industry clusters is challenging - and it is much better to just improve transportation toward them. Have you considered in the models the cost of moving enterprises or whole industries toward newly-built HSR areas? Do these costs (theoretically or practically) outweigh the benefits of being near the new HSR?

I generally like the flowcharts.

Are there hukou considerations here that add friction to your model, at least on the labor migration side?

Part 3.1 - you talk about method selection and solving problems without first telling the reader what those problems were and what the method is trying to do. Need at least 1-2 paragraphs before 3.1 to do that.

For the model (3.1), why not consider a two-way fixed effects version of the DID? That gets you year fixed effects along with the county fixed effects you already put in (ui). 

The DID model framework has seen much methodological work and critique in the past several years, especially in cases of variable treatment timing (which you have in your model, since different counties have HSR opened at different times). See Goodman-Bacon 2021 (https://www.sciencedirect.com/science/article/abs/pii/S0304407621001445), and Callaway and Sant'Anna 2022 (https://www.sciencedirect.com/science/article/abs/pii/S0304407620303948), for example. There are now different statistical packages to test for these issues in Stata, R, etc.

For the PSM, it would be best to see a table of how the variables used in the PSM balance between the treatment and control groups, before and after PSM. You state that PSM made things better and the Kernel Density charts show that indeed it is somewhat better, but showing the underlying PSM results would be better (could be appendix).

Regression tables: relabel the column headers so that reader actually knows which one is which.

Magnitudes: units are not considered anywhere. How should reader interpret a -.0001 log point change: is this economically meaningful or not? 

In the lagged regressions, why are the sample sizes decreasing so much - is it because the rail openings are so recent (<5 years) or lack of data?

Table 4 and 5 effects might be best shown as charts.

The discussion in 4.1.1 and 4.1.2 focuses its attention in proving out the superiority of PSM-DID to DID or non-DID in your application. I'm not sure why readers would assume otherwise. Your section on PSM (despite lack of balancing table) shows that PSM would likely be a better match. Why spend all the time comparing PSM-DID to others, instead of discussing your relatively interesting results of PSM-DID. And then do it again for the second dependent variable. Doesn't make sense to focus on that.

The multi-phase regressions need more explanation of the method. That's not the most standard version of DID and readers may be unfamiliar. Some citations to it would help. In fact, section 5.1 maybe should go up further in the paper.

The "lying city" concept needs explanation. What is it? It's not introduced until the conclusion (but appears in abstract!) and not clear what authors mean by it.

Paper needs proofreading and contains some non-standard English usage and punctuation choices. There are many fragments & incomplete sentences.

Author Response

We would like to thank you for your thoughtful review, understanding of the language and expression being underdeveloped, recognition of the research. The valuable and constructive feedback that addressed several research limitations, which has significantly improved the presentation, the clarity, coherence, and persuasiveness of our manuscript.During the process of revising our work, we carefully reconsidered our language, theoretical frameworks, Concept interpretation, research design method selection, research conclusion deficiency and content presentation method improvement. Additionally, during the revision process, we can learn various techniques and gain insights from cutting-edge literature that will enhance the quality of this article. The explanation of what we have changed in response to the suggestions is given point by point in the following pages, please see the attachment.  

Round 2

Reviewer 2 Report

This is a MUCH improved version. I commend the authors for a thorough rework of most of the paper.

A few more revisions suggested:

p.7, first paragraph, says "Here, we propose hypothesis 2: introducing high-speed rail will have a long-lasting and good impact on the county's secondary labor force concentration". I would rephrase to "positive impact" or "increase the county's secondary labor force concentration". 'Good' is not appropriate and seems like a value judgment.

Table 3 model sub-headers are still too jargony. what does "on_support" and on_support&weight!=." mean to the average reader? Please edit.

4.2.2. I like that you have these estimates over time + 95% CI graphs for sections 4.2.1. It really illuminates the effect. In Figure 6, the estimates are clean and the CI does not cross the 0 line, suggesting a statisticaly significant impact over time, with a clear trend. Figure 7 however, shows that the point estimates and the CIs are very near 0 and mostly contain zero. This suggests that these are not statistically significant differences at least over time. You should reword your conclusion in 4.2.2 to reflect that. I do not agree that Figure 7 results show the conclusions (1) and (2) that you reach in 4.2.2. Your findings in the regression table on the overall point estimates might be significant at the 5% level, but your findings over time do not seem that way. You need to acknowledge that and not just bury it.

Conclusion: again, the "lying cities" concept doesn't make sense without a definition. If this is a translation issue, then find a different translation. If you mean "bedroom communities" then use that. If it's something else, then suggest a definition / link to reference, etc.

Author Response

We would like to thank you for your thoughtful review again. The suggestions of what we have omitted and distorted have once more emphasized the research's scholarly and scientific nature. The explanation of what we have changed in response to the suggestions is given point by point in the following pages. Please see the attachment.
